# Microvariation at the Interfaces: The Subject of Predication of Broad Focus VS Constructions in Turinese and Milanese

**Delia Bentley** [1,*] and **Francesco Maria Ciconte** [2,*]

1   Department of Linguistics and English Language, The University of Manchester, Manchester M13 9PL, UK
2   Department of Humanities, Arts and Social Sciences, University of Chieti-Pescara, 66100 Chieti, Italy
*   Correspondence: delia.bentley@manchester.ac.uk (D.B.); francesco.ciconte@unich.it (F.M.C.)

**Abstract:** Presentational constructions, i.e., structures which introduce an event into the universe of discourse, raise the question of what it means for a predication to be entirely new in information structural terms. While there is growing consensus that these constructions are not topicless, there is no agreement on how to analyse their topic. The Romance languages of Northern Italy have figured prominently in this debate because the presentational constructions of many such languages exhibit VS order and an etymologically locative clitic in subject clitic position. This clitic has been claimed to be a subject of predication in a syntactic subject position. Adducing primary comparative evidence from Milanese and Turinese, we discuss patterns of microvariation which suggest that the etymologically locative clitic need not be a syntactic subject and can mark an aboutness topic provided by the discourse situation alone. We propose a parallel-architecture, Role and Reference Grammar account whereby the microvariation under scrutiny is captured in terms of the interfaces that are involved in the parsing of utterances. This account considers discourse to be an independent module of grammar, which, alongside the semantic and syntactic modules, is directly involved in linguistic variation and change.

**Keywords:** aboutness topic; interfaces; microvariation; parallel architecture; presentational construction; Role and Reference Grammar; Romance; subject clitic



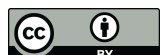

## 1. Introduction

Many dialects of Northern Italy exhibit an etymologically locative clitic in sentences with VS order which introduce a new event into the universe of discourse (Tortora 1997, 2014; Parry 2000, 2013; Manzini and Savoia 2005, vol. 2; Ciconte 2008, 2011; Pescarini 2016, p. 749; Bentley 2018; Flecchia 2021, 2022; Bentley and Cennamo 2022).[1] We call these sentences *presentational constructions* (Parry 2013, p. 511), and we refer to the etymologically locative clitic as *presentational clitic*. Milanese *ghe* is a relevant example.[2]

(1)  **Gh'**      è        rivà           i       to      surèi.    (Milanese)
     PRESCL    be.3SG   arrive.PSTP    the     your    sisters
     'Your sisters have arrived.'

The presentational clitic occurs not only with verbs of motion, like *rivàr* 'arrive' in (1), but also with verbs that lack a locative argument, like *murìr* 'die' in (2).

(2)  **Gh'**      è        mort          tanti    suldà.    (Milanese)
     PRESCL    be.3SG   die.PSTP      many     soldiers
     'There died many soldiers.'

This suggests that rather than encoding a location, *ghe* signals a property of presentational focus, which deserves investigation. Comparing the development of the presentational clitic to that of the existential proform 'there' (Ciconte 2008, 2011), the existing analyses argue that the presentational clitic, originally a resumptive locative pronoun, developed into a subject agreement marker, i.e., the marker of verbal agreement with a

covert locative subject of predication (see Parry 2013 for the diachronic account; for the clitic in synchronic terms, see Burzio 1986; Saccon 1992; Parry 1997, p. 243; 2000; Tortora 1997; 2014, pp. 29–32). The subject of predication is understood as a spatio–temporal location, which is required by the presentational construction (Benincà 1988; Calabrese 1992; Saccon 1992). The analyses of the clitic as a subject agreement marker are supported by the observation that the presentational clitic takes the position of a subject clitic, i.e., a bound pronominal form, which is an extended exponent of subject–verb agreement. In fact, the presentational clitic clusters with or occurs in complementary distribution with subject clitics.[3] The verb also fails to agree in number with the postverbal NP, which suggests that the latter is not the subject of the presentational construction: see the number mismatch between the singular auxiliary *è* 'is' and the plural NPs *i to surèi* 'your sisters' and *tanti suldà* 'many soldiers' in (1) and (2), respectively.

Comparative evidence from other dialects of Northern Italy reveals, however, that there is variation in the position of the presentational clitic and in the agreement relation between the verb and the postverbal NP. Thus, Turinese *je* contrasts with Milanese *ghe*, in that it does not figure in a subject–clitic position (cf. 3 and 4) and it is not ruled out when the verb agrees in number with the postverbal NP (cf. 4).[4]

| (3) | A | l′ | è | riva**je** | | toe | sorele. | (Turinese) |
|---|---|---|---|---|---|---|---|---|
| | SCL.3SG | AUXCL | be.3SG | arrive.PSPT.PRESCL | | your | sisters | |
| | 'Your sisters have arrived (here/where I am/was).' | | | | | | | |
| (4) | A | **son** | rivà**je** | | toe | sorele. | | (Turinese) |
| | SCL.3PL | be.3PL | arrive.PSTP.PRESCL | | your | sisters | | |
| | 'Your sisters have arrived (here/where I am/was).' | | | | | | | |

The evidence in (3) and (4) challenges the view that the presentational clitic satisfies a syntactic subjecthood requirement. However, it does not conflict with an analysis of the clitic as the expression of the deixis of the discourse situation in which the new event is announced and to which it is relevant. In such cases, the discourse situation is the aboutness topic of the utterance, or what the utterance is about (Gundel 1988), and the presentation clitic indexes it by virtue of its deictic features. Supported by the speaker-oriented deixis of presentational constructions with *je* (see the translation of examples 3 and 4), this hypothesis draws on the notion that, despite introducing a new event, apparently topicless sentences require a stage topic (Erteschik-Shir 1997, p. 8) or, otherwise put, a subject of predication (Saccon 1992; Benincà 1988; Calabrese 1992; Bianchi 1993; Parry 2013).

In this article, we examine first-hand Turinese and Milanese evidence collected in loco with questionnaire-assisted interviews. We claim that the microvariation in the presentational constructions of the two dialects resides at the interfaces that are involved in sentence parsing. We frame our account in terms of Van Valin's (2023, pp. 123–25) syntax–semantics linking algorithm, which is an idealization of the hearer's perspective in linguistic communication. The presentational clitic can be understood as a locative argument of a verb of motion, in which case it is linked from syntax to a position in the semantic representation of the clause. Alternatively, the clitic indexes the discourse situation and is directly linked from syntax to discourse. This latter interpretation is not restricted to presentational constructions with verbs of motion. To capture the variation attested in Turinese (cf. 3 and 4), we advance the hypothesis that a new process of grammaticalization is under way, where referential, i.e., locative, *je*, which was never ousted from the system, is being reanalysed for a second time. As a result of the reanalysis, *je* becomes an index of the aboutness topic of the utterance, which is the discourse situation.

Both the construal of *je* as a locative argument and that as an index of the discourse situation are compatible with the establishment of an agreement relation between the verb and the postverbal NP, the notion of subjecthood being broken down, in our analysis, into an aboutness relation, which is not in principle syntactic and can be satisfied in discourse, and an agreement relation, which is syntactic and construction-specific (Van Valin and LaPolla 1997, pp. 242–309; LaPolla 2023). The construction-specific agreement relation is missing obligatorily in the Milanese presentational construction with the presentational

clitic (cf. 1 and 2), and optionally in its Turinese counterpart (cf. 3 and 4). In the absence of V–S agreement, the grammars of both dialects require that the aboutness relation be expressed overtly by the presentational clitic (cf. 1–3). We take this to be a constructional requirement of the autochthonous presentational pattern of these dialects. Instead, the presentational construction with V–S agreement is a result of contact with and pressure from Italian.

In the sections to follow, we discuss the micro-typology of presentational constructions found in Milanese and Turinese (Section 2). We then briefly introduce Parry's (2013) analysis of the diachrony of presentational clitics (Section 3.1), and we advance a hypothesis on the variation observed in Turinese nowadays (Section 3.2). We introduce our framework in Section 4, and we propose our formal analysis in Section 5. Brief conclusions are drawn in Section 6.

## 2. Microvariation in the Presentational Construction

The Milanese and Turinese clitics introduced in the previous section illustrate two of the three formal types of presentational clitic attested in the dialects of Northern Italy: proclitic (cf. 1 and 2), enclitic (cf. 3 and 4), and sequential (cf. 5). This micro-typology is sketched in Table 1.

| (5) | **Ngh**′ | è | gnö | denti-**ghi** | na | segretaria | (Borgomanero) |
|-----|----------|---|-----|---------------|----|-----------|---------------|
|     | LOC      | be.3SG | come.PSTP | inside-LOC | a | secretary | |
|     | int      | la | stônza | | | | |
|     | in       | the | room | | | | |

'A secretary entered the room.'
(Tortora 2014, p. 20)

**Table 1.** A micro-typology of presentational clitics.

| Type I: Proclitic | Type II: Enclitic | Type III: Sequential |
|-------------------|-------------------|----------------------|
| Milanese *ghe* (cf. 1 and 2) | Turinese *je* (cf. 3 and 4) | Borgomanerese *ngh'… gghi* (cf. 5) |

An etymologically locative clitic is known to have undergone partial univerbation in some of the Northern dialects, figuring on the inflected forms of 'have' and 'be' under conditions outlined in Benincà (2007).

| (6) | **Gh'ò** | cantà. | (Venetan) |
|-----|----------|--------|-----------|
|     | CL-have.1SG | sing.PSTP | |

'I have sung.'
(Benincà 2007, p. 28)

Given that our focus is on presentational constructions, we shall leave this last form out of the discussion. In what follows, we introduce our findings on Milanese and Turinese, which exhibit clitics of types I and II.

### 2.1. Our Findings

We conducted questionnaire-assisted interviews with Milanese and Turinese speakers and found that, in both dialects, the absence of agreement is normally dependent on two conditions: (i) S must not be a personal pronoun, and (ii) V must be a Vendlerian state, achievement, or accomplishment.[5] Thus, the second person plural pronoun obligatorily controls person and number agreement on the verb in (7) and (8).[6]

| (7) | Dopu | si | / | *gh | / | *l′ | *è | rivà | vuialter. | (Milanese) |
|-----|------|----|----|-----|----|-----|-----|------|-----------|------------|
|     | after | be.2PL | | PRESCL | | AUXCL | 3SG | arrive.PSTP | you.PL | |

'Then you arrived.'

(8) Peui     i          seve       vnü            /    *a        *l'è            (Turinese)
after   SCL.2PL   be.2PL   come.PSTP           SCL.3SG   AUXCL-be.3SG
vnü(je)                          voi.
come.PSTP(PRESCL)                you.PL
'Then you came.'

In (9) and (10), V–S agreement is, instead, required because the verb *balé* 'dance' is a Vendlerian activity (see Parry 2013; Bentley 2018; Bentley and Cennamo 2022 for further detail).

(9)   (A     la      festa)    an         balà       /   *l'      /   *gh'    a          (Milanese)
at     the     party    have.3PL   dance.PSTP   AUXCL       PRESCL have.3SG
balà           i        to         gent.
dance.PSTP     the      your       parents
'(At the party) your parents danced.'

(10)  (A     la      festa)     a          l'an                balà          /    *a         (Turinese)
at     the     party     SCL.3PL    AUXCL-have.3PL      dance.PSTP          SCL.3SG
l'a                     balà(*je)                tò      papà   e      toa    mama.
AUXCL-have.3SG          dance.PSTP(PRESCL)        your    dad    and    your   mum
'(At the party) your mum and dad danced.'

The second condition appears to be sporadically violated in Turinese, a point to which we return in Section 3.2.

Apart from these shared constraints, in Milanese we found two patterns. The one illustrated in (11) was prevalent: the auxiliary hosts the presentational clitic and does not agree in number with the postverbal noun phrase.

(11)  a.   *(**Gh'**)è             rivà         i      to       surèi   /    di     pac.    (Milanese)
PRESCL-be.3SG       arrive.PSTP   the   your     sisters   of     parcels
'Your sisters have arrived'/'There arrived some parcels.'

b.   *(**Gh'**)è             mort         tanti     suldà.
PRESCL-be.3SG       die.PSTP     many      soldiers
'There died many soldiers.'

V–S agreement was also attested, but the presentational clitic turned out to be incompatible with the agreeing verb.

(12)  a.   (*Gh')in              rivà         i      to       surèi   /   di    pac.    (Milanese)
PRESCL-be.3PL       arrive.PSTP   the   your     sisters   of    parcels
'Your sisters have arrived.'/'There arrived some parcels.'

b    (*Gh')in              mort         tant   sulda.
PRESCL-be.PL        die.PSTP     many soldiers
'There died many soldiers.'

Subject clitics are known to be incompatible with preverbal object clitics in some Friulian and Francoprovençal varieties (Benincà and Vanelli 1986; Roberts 1993; Poletto and Tortora 2016, p. 785). We found that the presentational clitic could not occur in the presence of a reflexive clitic, and, in this case, the verb obligatorily agreed in number with the postverbal NP.

(13)  Varda:       s'in          s-cepà       tanti    ram.               (Milanese)
look.IMP.2SG   REFL-be.3PL   break.PSTP   many     tree-branches
'Look! Many branches have broken.'

In sum, the Milanese presentational clitic *ghe* is required in VS presentationals without verb agreement and is incompatible with such agreement. Taking V–S agreement and the presence of the presentational clitic to be two binary variables, which we call {±Agr} and {±Cl}, the situation found in Milanese presentational constructions can be represented as in Table 2.

**Table 2.** Milanese presentational constructions: two patterns.

|  | Pattern (i) | Pattern (ii) |
|---|---|---|
| {Agr} | − | + |
| {Cl} | + | − |

At this juncture, a brief digression is necessary. We should note that Milanese *ghe* does co-occur with the inflected copula of locative and existential constructions exhibiting a postcopular personal pronoun, as we illustrate here.

(14) Chi l'è che gh'è in cüsina? (Milanese)
who AUXCL-be.3SG that LCL-be.3SG in kitchen
**Ghe** sun mi, in cüsina.
LCL be.1SG I in kitchen
'Who is the kitchen? I am the kitchen (lit., There am I, in the kitchen).'

(15) Maria l'è no in de per lé: (Milanese)
Mary SCL-be.3SG NEG in by for her
**ghe** sun mi.
PF be.1SG I
'Mary is not alone: I am there for her (lit., There am I).'

The *there* form of locatives and existentials must, however, be distinguished from the presentational clitic on both empirical and theoretical grounds. First, both in Italian and in some northern dialects of Italy not discussed in depth here, an etymologically locative clitic occurs in locatives and existentials (cf. 16 and 17) but is unattested in presentationals (cf. 18).

(16) a. (In cusina) **gh** son mi. (Grosio)
in kitchen LCL be.1SG I
'I am in the kitchen (lit., In the kitchen there am I).'

b. (In cucina) **ci** sono io. (Italian)
in kitchen LCL be.3PL I
'I am in the kitchen (lit., In the kitchen there am I).'

(17) a. Maria l'é miga de per lé: (Grosio)
Mary SCL-be.3SG NEG by for her
**ghe** son mi.
PF be.1SG I
'Mary is not alone: I am there for her (lit., There am I).'

b. Maria non è sola: **ci** sono io. (Italian)
Mary NEG be.3SG alone PF be.1SG I
'Mary is not alone: I am there for her (lit., There am I).'

(18) a. L'é rivä i toa sureli. (Grosio)
SCL-be.3SG arrive.PSTP the your sisters
'Your sisters have arrived.'

b. Sono arrivate le tue sorelle. (Italian)
be.3PL arrive.PSTP.PL the your sisters
'Your sisters have arrived.'

While the proform in (16) resumes the previously introduced locative predicate *in cusina* 'in the kitchen', hence the gloss LCL 'locative clitic', the existential proform in (17), glossed PF, has been argued to signal the context dependence of existential sentences, which are predications of an implicit contextual argument (Bentley et al. 2015, p. 146; following Francez 2007). In contrast, the analysis of the presentational clitic is not similarly straightforward. The presentational construction is fully interpretable in its absence and is truth-conditionally equivalent with the corresponding SV sentence (Lambrecht 1988, p. 115;

Karssenberg 2016; 2018a, p. 23; 2018b). Furthermore, depending on the semantics of the verb, the presentational clitic can receive a locative interpretation (Section 1). In the present work, we shall, therefore, leave aside existentials and locatives and focus on presentational constructions, as defined in Section 1, because it is in such constructions that ambiguity arises in the interpretation of the etymologically locative clitic, revealing the key role of the interfaces in microvariation.

Turning now to Turinese, to begin with, not one but two etymologically locative forms are found in the presentational constructions of this dialect (Parry 1997, 2000, 2010, 2013). At first, the two forms appear to be allomorphs of the same morpheme (see, e.g., Parry 2013, p. 514): the one, *j*, occurs proclitically to the finite verb in the simple tenses (cf. 19), whereas the other, *je*, occurs enclitically to the participle (cf. 20) or the infinitive.

(19)  Se        a-**j**           seurt              ël     sol,   sì      a-**j**
      if        EXPL-PRESCL       come.out.3SG       the    sun    here    EXPL-PRESCL
      nass      ji                bolè.[7]
      be.born.3SG  the            mushrooms
      'If the sun comes out, mushrooms will appear here.'
      (Parry 2013, p. 515, data from Burzio 1986)

(20)  Che       bel!        A          l'è              na(ssù)**je**        (Turinese)
      what      beautiful   SCL.3SG    AUXCL-be.3SG     be.born.PSTP.PRESCL
      le        fior.
      the       flowers
      'How nice! The flowers have appeared.'

On further inspection, this hypothesis would appear to be problematic. Unlike *je*, *j* is not constrained in terms of the Aktionsart of the verb and, when occurring with activity verbs, it is associated with habitual or iterative aspect (Parry 2013, p. 541).

(21)  An        cost        let    a-**j**           deurm      mie    fije.    (Turinese)
      in        this        bed    EXPL-PRESCL       sleep.3SG  my     daughters
      'This bed is where my daughters sleep.'
      (Parry 2013, p. 541)

In addition, the pattern with *j* was not normally chosen by our informants. When it was chosen, *j* was hosted by an inflected form of the verb (for the latter point, see also Tosco et al. 2023, p. 184).

(22)  Guardoma  la      partita    e        a(-**j**)              intro      (Turinese)
      watch.1PL  the    game       and      SCL.3PL(-PRESCL)       enter.3PL
      doi       lader   dal        giardin.
      two       thieves from-the   garden
      'We are watching the game and two thieves enter from the garden.'

The evidence in (22) contrasts with that reported in the literature (cf. 19, 21), including the treatments of earlier stages of the language (Parry 2013, p. 539), where *j* correlates with lack of V–S agreement.

In contrast with *j*, *je* is very well attested in our dataset, and its distribution vis-à-vis V–S agreement is not the same as that found with *j*. In fact, we found the threefold possibility illustrated in Table 3. The fourth logical combination of the two variables, the lack of both V–S agreement and *je*, was not attested in Turinese.

**Table 3.** Turinese presentational constructions with *je*: three patterns.

|         | **Pattern (i)** | **Pattern (ii)** | **Pattern (iii)** |
|---------|:---------------:|:----------------:|:-----------------:|
| {Agr}   | −               | +                | +                 |
| {Cl}    | +               | −                | +                 |

Patterns (i) and (ii) of Table 3 are the same as those found with *ghe* (see Table 2), whereas according to our records, pattern (iii) is unknown to Milanese. The three combinations of the two binary variables are illustrated in the following examples from Turinese.

[pattern (i)]

(23)　a.　A　　　　　　l'è　　　　　　　　riva**je**　　　　　　toe　　　sorele
　　　　　　SCL.3SG　　AUXCL-be.3SG　arrive.PSTP.PRESCL　your　　sisters
　　　　　　di　　　　　　pachet.
　　　　　　of　　　　　　parcels
　　　　　　'Your sisters have arrived'/'There arrived some parcels.'

　　　b.　A　　　　　　l'è　　　　　　　　nassù**je**　　　　　　　tante　　fior.
　　　　　　SCL.3SG　　AUXCL-be.3SG　be.born.PSTP.PRESCL　many　　flowers
　　　　　　'Many flowers have appeared.'

[pattern (ii)]

(24)　a.　A　　　　son　　　montà　　　　ën　　　país　　i　　　tòi
　　　　　　SCL.3PL　be.3PL　go.up.PSTP　in　　　village　the　　your
　　　　　　nòno.
　　　　　　grandparents
　　　　　　'Your grandparents have gone/come up to the village.'

　　　b.　A　　　　son　　　calà　　　　　　　i　　　sgnor　　dël
　　　　　　SCL.3PL　be.3PL　come.down.PSTP　the　　people　　of.the
　　　　　　pian　　ëdzora.
　　　　　　floor　　of.upstairs
　　　　　　'The people from the upstairs floor have come down.'

[pattern (iii)]

(25)　a.　A　　　　　**son**　　rivà**je**　　　　　　toe　　　sorele　　/
　　　　　　SCL.3PL　be.3PL　arrive.PSTP.PRESCL　your　　sisters
　　　　　　di　　　　　　pachet.
　　　　　　of　　　　　　parcels
　　　　　　'Your sisters have arrived.'/'There arrived some parcels (here, where I am).'

　　　b.　A　　　　　**son**　　nassu**je**　　　　　　tante　　fior.
　　　　　　SCL.3PL　be.3PL　be.born.PSTP.PRESCL　many　　flowers
　　　　　　'Many flowers have appeared.'

Three observations on the patterns in (23)–(25) are in order. Firstly, according to the literature, pattern (i) is the autochthonous Turinese presentational construction (Parry 1997, p. 243; 2013, pp. 514–15; Flecchia 2022, p. 44). Pattern (ii), instead, is the outcome of convergence between the grammar of Turinese and that of the more prestigious language Italian (Flecchia 2022), which has no presentational clitic and requires V–S agreement in presentationals (cf. 18b).

Secondly, *je* occurs in complementary distribution with any non-subject clitics that the sentence may require. In (26), we provide an example with the direct object clitic *lo* 'it'; the position of *je* is the same as that of this clitic. In (27) we report a presentational construction, where the participle hosts a dative and a partitive clitic. *Je* is banned with consequent obligatoriness of V–S number agreement.

(26)　Col　　film,　　a　　　　　l'an　　　　　　　vist-lo　　　　tuti　　(Turinese)
　　　　that　　film　　SCL.3PL　AUXCL-have.3PL　see.PSTP-OCL　all
　　　　i　　　mè　　amis.
　　　　the　　my　　friends
　　　　'That film, all my friends have seen it.'

(27)　A　　　　　son　　　rivà-(*je)-m-ne(*-je)　　　　　　　　　doi.　(Turinese)
　　　　SCL.3PL　be.3PL　arrive.PSTP(*PRESCL).DATCL.PARTCL(*PRESCL)　two
　　　　'There arrived two of them to me.'

Thirdly, according to many informants *je* has a speaker-oriented deictic function, indicating that the location of the event is the same as that of the speaker, and suggesting first-hand witnessing of the event.

In sum, abstracting away from the shared constraints cited at the beginning of this section (cf. 7–10), a more varied array of patterns was found in the presentational constructions of Turinese than in those of Milanese. In the latter dialect, the presentational proclitic *ghe* is hosted by a verb that will exhibit invariant third-person singular morphology, regardless of the number feature of the postverbal NP. Although V–S number agreement is also an option, it is mutually exclusive with the presentational clitic. As for Turinese, in contrast with what is reported in the literature, we found the proclitic *j* to occur rarely in presentational constructions and to have no effect on the number agreement between the verb and the postverbal NP, which is regularly found in the present. The other form known to occur in Turinese, *je*, attaches enclitically to the participle of the perfect, on a par with non-subject clitics. While the literature reports that *je*, like Milanese *ghe*, is incompatible with V–S agreement, we found this not to be the case: *je* and the number agreement specifications on the perfect auxiliary are not mutually exclusive.

## 3. The Development of the Presentational Clitic and Its Theoretical Consequences

### 3.1. The Diachrony of J: Parry's Account

The development of the presentational clitic is well documented and has been studied in some depth. An etymologically locative form is attested in early Italo–Romance existentials since the 12th century, its reanalysis into a non-referential existential proform dating from the 13th–14th centuries (Ciconte 2008, 2011, 2013). To capture this development, Parry (2013) has claimed that the existential proform, which was not found in Latin existentials, originated as a locative pronoun which resumed an extra-clausal locative phrase or a locative phrase that was interpolated between the verb and its postverbal argument: see the [V-Loc-NP] order in (28).

(13th c. Veronese, Giacomino da Verona, *Babilonia*)

(28)      Asai      **g**′è      **là**      **çó**      bisse [...]
         many      there-is      there      down      grass-snakes
         'There are many grass-snakes [...] down there.'
         (Parry 2013, p. 530)

In the V2 syntax of old Romance, which was characterised by (X)VS order, the postverbal position was the default position of the subject. Therefore, in the structures where the locative phrase was interpolated between the verb and its postverbal argument, the locative phrase lent itself to being analysed as the subject. In turn, the co-referring clitic could be reanalysed as a subject agreement marker. According to Parry, the existential proform originated from this reanalysis.

In Piedmontese, the presentational clitic J (*j*/*je*) did not spread to presentational constructions until the 17th century, and even then, it was only found with the verb *arivé* 'arrive' (Parry 2013, p. 539). By the 18th century, it was attested with a larger range of verbs of directed motion and change of state, as can be seen below.

(18th c. Turinese, I. Isler, ed. Viglongo 1968)

(29)      L'    é    bin    dal    liam    ch'    a**i**          nass
         it    is    well    from.the    manure    that    EXPL.SCL-LOC.CL    be.born.3SG
         le    fior.[8]
         the    flowers
         'Indeed, it's from manure that flowers grow.'
         (Parry 2013, p. 539)

According to Parry, in this structure a referential locative clitic was also reanalysed as a subject agreement marker: "Indeed, it was a similar process of syntactic reanalysis that produced subject–clitic agreement markers. The latter were originally used as clause–internal resumptive pronouns linked to dislocated subjects, but later weakened into compulsory

agreement markers on the verb (see for example, Poletto 1993, and for Piedmontese, Parry 1993)." (Parry 2013, p. 529).

This reanalysis was favoured by the postverbal occurrence of the argument (see *le fior* 'the flowers' in 29), and the failure of the verb to enter into an agreement relation with it (*nass* is singular in 29). Parry's claim is, therefore, that the locative clitic was reanalysed as an agreement marker, on a par with subject clitics. The presentational clitic thus came to be associated with a subject position, specifically, the position which Cardinaletti (2004) calls SubjP (Parry 2013, pp. 535–36), retaining some locative value at the level of discourse and spelling out the locative subject of predication of the presentational construction.[9]

### 3.2. The Re-Grammaticalization of je

At first, the evidence uncovered in our recent survey of Turinese appears to jar with the notion that a locative pronoun developed into a subject marker J, with allomorphs *j/je*. The challenge for this hypothesis is twofold. On the one hand, we found that both *j* and *je* occur in VS constructions where the verb patently agrees in number with the postverbal NP (cf. 22, 25a,b). This fact is irreconcilable with the assumption that *j* and *je* signal that the verb agrees with a subject of predication that is different from the postverbal NP. On the other hand, *j* hardly occurs in our dataset, whereas *je* figures not only in the autochthonous pattern known from the literature (cf. 23a,b), but also in a presentational pattern with V–S agreement, which seems to have gone unnoticed so far (cf. 25a,b).

Capitalising on the observation that patterns (ii) and (iii) from Table 3 (cf. 24, 25a,b) were not traditionally found in Turinese, we propose that the new evidence gathered in our survey does not challenge Parry's (2013) hypothesis on the development of a locative pronoun into a presentational clitic, but rather suggests that new variation is available in contemporary Turinese grammar, and a new development may be under way. Starting from the outcome of the diachronic development discussed in Parry (2013), i.e., pattern (i) from Table 3 (cf. 30a), a new presentational pattern was introduced into the system because of contact with Italian (Ricca 2008; Flecchia 2022), namely pattern (ii) (cf. 30b). Our hypothesis is that J had never lost its locative meaning, but rather had undergone layering, maintaining its old function at the same time as the presentational one (for layering, see Hopper and Traugott 1993, pp. 36, 124–26; for locative j, see Tosco et al. 2023, p. 184). Locative J was introduced into pattern (ii) as the spell-out of a locative argument or to resume a locative adjunct. This is how pattern (iii) originated.

(30)
a. **Stage 1**: Pattern i {+J; −V–S Agr}.
b. **Stage 2**: Pattern i {+J; −V–S Agr}; pattern ii {−J; +V–S Agr}.
c. **Stage 3**: Pattern i {+J; −V–S Agr}; pattern ii {−J; +V–S Agr}; pattern iii {+J; +V–S Agr}.

That J can be locative is suggested not only by the speakers' observation that *je* can have a deictic flavour, signalling the location of the speaker, but also by the few exceptions to the Aktionsart constraint discussed in Section 2.1 (cf. 10). According to our informants, such violations of the ban on activity predicates in presentational focus are only allowed if the location of the event is specified, as can be seen in (31). This structure was provided spontaneously by several informants.

| (31) | **A** | **la** | **scola**, | a | l'an | durmì**je** | (Turinese) |
|---|---|---|---|---|---|---|---|
| | at | the | school | SCL.3PL | AUXCL-have.3PL | sleep.PSTP-LCL | |
| | tanti | cit. | | | | | |
| | many | children | | | | | |

'At school, many children have slept (there) (lit. there slept many children).'

The locative interpretation also arises naturally from examples like (25a), repeated in (32) for convenience, where the verb describes directed motion.

(32)     **A**       **son**       rivà**je**       toe     sorele    /      (Turinese)
       SCL.3PL    be.3PL    arrive.PSTP.PRESCL    your    sisters
       di        pachet.
       of       parcels
       'Your sisters have arrived.' / 'There arrived some parcels (here, where I am).'

However, pattern (iii) is also attested, if more rarely, with verbs that do not take a locative argument, like *nasse* 'be born' and *dimagrì* 'lose weight'.

(33)    a.      A       son      nassu**je**       tante    fior.    (Turinese)
           SCL.3PL    be.3PL    be.born.PSTP-PRESCL    many    flowers
           'Many flowers have appeared.'

        b.      A       son      dimagrì**je**       tanti    cit.
           SCL.3PL    be.3PL    lose.weight.PSTP.PRESCL    many    children
           'Many children have lost weight.'

This suggests that the locative pronoun of pattern (iii) is subject to reanalysis in synchrony and, as a consequence, is being re-grammaticalized into the marker of a more abstract property of the construction, which is only loosely related to locative meaning. To understand what this property might be, it is important to return to the point made previously that *j*, the allomorph of J expected to figure in the simple tenses, makes only a few sporadic appearances in our dataset. In other words, in the examples in the present tense, pattern (ii) was normally chosen. The optionality of *j* was shown in (22); in (34), we report the present-tense pattern that was by far predominant in the responses to our questionnaire.

(34)    a.      A-j       è      la      guera:               (Turinese)
           SCL.3SG-PF    be.3SG    the     war
           **a**        **moero**    tanti    soldà.
           SCL.3PL    die.3PL    many    soldiers
           'There's a war: many soldiers are dying.'

        b      St'ane     sì      **a**       **naso**      poche    masnà.
           this-year    here    SCL.3PL    be.born.3PL    few    children
           'Only few children have been born this year.'

From the contrast between the present and the perfect, we conclude that the erstwhile allomorphs of J have parted ways, with *j* becoming virtually obsolete, and *je* acquiring a new function of its own, a function which emerges in a construction that requires the perfect, where *je* was always licensed, namely the presentational construction. More specifically, *je* is reanalysed as the marker of the deixis of the discourse situation in which the announcement of a new event is made.[10] Since it is compatible with the agreement specifications on the verb, it must be concluded that it is not a syntactic subject. However, in light of the obligatoriness of *je* in the absence of agreement, it must be the case that the presentational clitic is a constructional requirement in Turinese (as in Milanese), when the construction is unambiguously presentational.[11]

Since we assume that the last stage in the development outlined in (30) involves a new role for *je*, which has parted from *j*, we represent the variation attested in contemporary Turinese as follows.

(35)

Pattern (i) {+*je*; −V–S Agr} ~ pattern (ii) {−*je*; +V–S Agr}~ pattern (iii) {+*je*; +V–S Agr}

After introducing the framework adopted in our analysis (Section 4), in Section 5 we shall provide a formal account of this variation, outlining the synchronic conditions for the reanalysis of *je*.

### 4. Role and Reference Grammar

Role and Reference Grammar (henceforth RRG; see Foley and Van Valin 1984; Van Valin and LaPolla 1997; Van Valin 2005, 2023; Bentley et al. 2023) is a *parallel architecture* theory (Jackendoff 2002, pp. 125–30), which represents discourse–pragmatics and semantics separately from syntax and seeks explanation in the interplay of these independent modules of grammar. A bidirectional algorithm governs the mapping—or *linking*—of semantics with syntax, in language production, and syntax with semantics, in language comprehension. Regardless of direction, the linking consists of two phases: the lexical phase builds the meaning of the clause, starting from the lexical meaning of the predicators, and assigns macroroles, i.e., generalized semantic roles, to the arguments, following universal principles that are grounded in Dowtyan lexical decomposition rules (Van Valin and LaPolla 1997, pp. 139–78; Van Valin 2023, p. 113). The morphosyntactic phase determines the morphosyntax of the clause and is characterized by a great deal of cross-linguistic variation to do with the assignment of grammatical relations, voice alternations, alignment, head- vs. dependent-marking orientation, etc. (Van Valin and LaPolla 1997, pp. 242–309; Van Valin 2023, pp. 147–49). The linking is paralleled by the discourse–pragmatic dimension, which can intervene at any point, although its role is most acutely felt in the morphosyntactic phase, resulting in significant intra- and cross-linguistic variation (Bentley 2023b). The relation between the three dimensions is represented schematically in Figure 1.

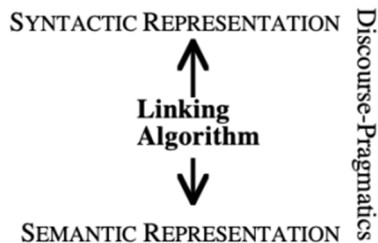

**Figure 1.** The interplay of syntax, semantics and discourse pragmatics (Van Valin 2023, p. 19).

Grammatical relations are not universals of syntactic theory for RRG; nor is there a subjecthood well-formedness requirement on the clause (LaPolla 2023). In the languages which do have grammatical relations, these are defined in terms of construction-specific restricted neutralisations of semantic roles for syntactic purposes. Thus, taking number verbal agreement to be a construction, this is a grammatical relation in the languages under discussion because it neutralises the distinction between A (actor of transitive), S (actor and undergoer of intransitive) and d-S (derived S in the passive), leaving out U (undergoer of transitive). Actor and undergoer are macroroles, or generalised semantic relations, in RRG, a point to which we return below. What matters here is that the contrast between the two is neutralized by number agreement but leaves out U, undergoer of transitive. The grammatical relation which gathers {A, S, d-S}, leaving out U, is called a P(rivileged) S(yntactic) A(rgument).

Albeit formed in accordance with the general principles of clause construction, tree structures are templatic and stored in language-specific inventories. A parser intervenes early in the syntax–semantics linking to output a tree structure for the input received (Van Valin 2023, pp. 116–25). Neither movement nor empty positions are allowed, and therefore, tree structures must represent the actual order of the elements in the clause, thus satisfying a principle which is normally referred to as the concreteness constraint.

An important property of the RRG theory of grammar is that it is both projectionist and constructional (Bentley 2023a). It is projectionist in the sense that it derives key aspects of the syntax of the clause from facets of lexical meaning. It is constructional insofar as it assumes that competence in a language includes the knowledge of its constructions. Therefore, alongside the Syntactic Inventory and the Lexicon, the grammar of that language will include an inventory of Constructional Schemas, which are constellations of instructions for the formation and the parsing of each of the constructions of that language. In due course,

this aspect of the RRG conception of grammar will become relevant to our discussion. The general organization of grammar discussed thus far is illustrated in Figure 2.

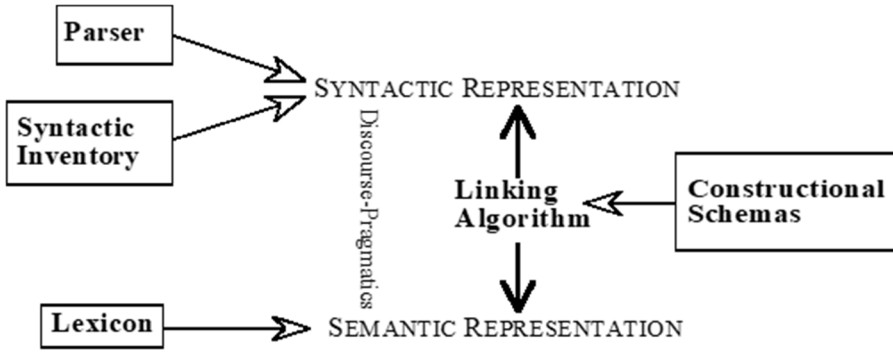

**Figure 2.** Organization of grammar in RRG (Van Valin 2005, p. 134).

A key principle governing the linking algorithm is the Completeness Constraint, which states that "[a]ll of the arguments explicitly specified in the semantic representation of a sentence must be realized syntactically in the sentence, and all of the referring expressions in the syntactic representation of a sentence must be linked to an argument position in [. . .] the semantic representation of the sentence." (Van Valin 2023, p. 116). While ensuring that every well-formed sentence is interpretable, this principle poses a challenge in cases of null anaphora (prop drop, object drop, silent predicates, etc.), since, as we said, the RRG representation of syntactic trees is constrained by a ban on phonologically null elements. Given that the modules of grammar can link directly with each other, RRG analyses null anaphora in terms of the direct linking of an argument or a predicate in the semantic representation of the clause with its representation in discourse without a concomitant link to syntax (Van Valin 2005; Shimojo 2008).

The RRG representation of discourse draws on von Heusinger's (1999) Discourse Representation Structures, which in turn build on Kamp and Reyle's (1993) Discourse Representation Theory. Discourse Representation Structures aim to capture the incrementality of information in discourse. They include variable-value pairs for the referring expressions that are gradually introduced into discourse and a representation of the semantic propositions in which the variables figure. As new propositions are gradually introduced into discourse, the co-reference relations between the variables of these propositions and those that were introduced previously are also represented. We shall provide examples of Discourse Representation Structures in Sections 5.1 and 5.2, although the issue of co-reference will not be relevant for our purposes, given than we deal with all-new utterances.

The direct linking of semantic and discourse representation that is assumed in RRG to capture null anaphora satisfies the *Extended* Completeness Constraint (Van Valin and Latrouite 2023, p. 496), which ensures that the sentence is fully interpretable without there being any null positions in syntax. The analysis which we shall develop in the next section will demonstrate another application of the Extended Completeness Constraint, whereby the direct linking connects discourse with syntax without involving the semantic representation built from the predicate(s) in the Lexicon.

## 5. Microvariation at the Interfaces

How can the synchronic variation discussed in previous sections be captured, and how is the change hypothesised for Turinese triggered and enabled? For ease of exposition, we illustrate the relevant variation again below.

(36)

Pattern (i) {+*je*; −V–S Agr} ~ pattern (ii) {−*je*; +V–S Agr} ~ pattern (iii) {+*je*; +V–S Agr}

As the reader will recall, patterns (i) and (ii) were found both in Milanese and in Turinese, although in the former dialect pattern (i) was predominant. The key difference

between the two dialects emerged instead from pattern (iii), which did not occur in Milanese but was well attested in Turinese. It is in this pattern, where the clitic co-occurs with V–S agreement, that the conditions for ambiguity in interpretation arise and it is this ambiguity that enables the new reanalysis of the clitic from locative to presentational (i.e., from a referential locative pronoun to the marker of the deixis of the discourse situation and hence an index of the aboutness topic of the utterance). The trigger of the reanalysis is the presentational construction itself because this construction does not provide an argument which can serve as an aboutness topic for the predication, but a topic is required by all propositions (Erteschik-Shir 1997). In what follows, we shall therefore analyse the two possible interpretations of the clitic and of pattern (iii), taking the perspective of the syntax–semantics linking, which is an idealization of the hearer's viewpoint in communication (Section 5.1). We shall then compare pattern (iii) with patterns (i) and (ii) (Section 5.2) and make some theoretical observations arising from this comparison (Section 5.3).

### 5.1. Pattern (iii) at the Syntax-Semantics Interface

Consider (37), which is a simplified version of (25a), and assume that this is an utterance that occurs out of the blue: there is no presupposition that x has arrived or that your sisters have done y.

| (37) | **A** | **son** | rivà**je** | | toe | sorele. | (Turinese) |
|------|-------|---------|------------|---|-----|---------|-----------|
| | SCL.3PL | be.3PL | arrive.PSTP.PRESCL | | your | sisters | |
| | 'Your sisters have arrived (here/where I am).' | | | | | | |

In accordance with the algorithm which governs the syntax–semantics interface (Van Valin 2023, pp. 123–25), once the input in (25) is received, the parser outputs a labelled tree structure (Step 1 in Figure 3). This will consist of a Nucleus, hosting the verb and the agreement specifications in the AG(reement) (Inde)X node, and R(eference) P(hrase)s for the pronoun *je* and the NP *toe sorele*. At this point, as much information is gleaned as is possible from the morphosyntax of the clause (Step 2). Given that the verbal inflection in the AGX node is in the third person plural, the clitic *a* is interpreted as a third person plural subject clitic (see note 4), and the plural NP *toe sorele* 'your sisters' is analysed as the controller of agreement or Privileged Syntactic Argument (PSA), which is the RRG name for a grammatical relation (Section 4). The morphosyntactic phase of the linking ends here.

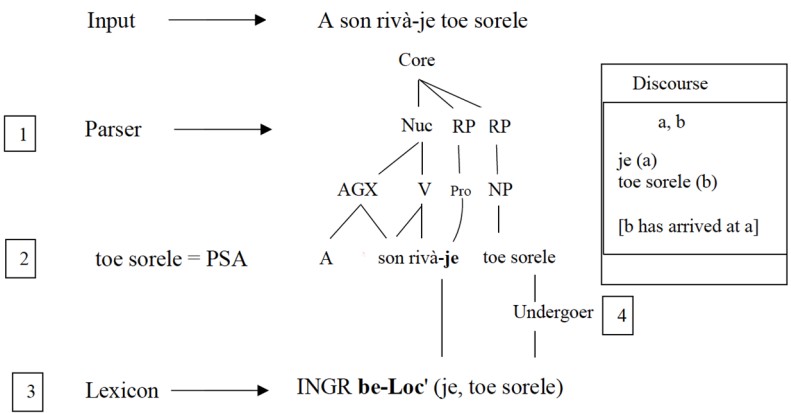

**Figure 3.** Pattern (iii) in syntax–semantics linking: locative interpretation of *je* in (37).

The lexical phase begins with the retrieval from the lexicon of the Logical Structure of the verb, i.e., the meaning representation with which it is stored. The RPs that have been introduced in discourse are assigned argument positions in this Logical Structure, following general lexical–decompositional principles (Van Valin and LaPolla 1997, pp. 113–16) (Step 3). The clitic *je* could in principle resume a locative adverbial that has been introduced into discourse previously or introduce anew a locative argument of the verb. Given that this is assumed to be an out-of-the blue utterance, *je* receives the latter interpretation. This is

represented in the Discourse Representation Structure, which parallels the syntax-semantics linking in Figure 3 (see the box called Discourse). This structure includes no presupposition and introduces anew the two variables a and b, their values *je* and *toe sorele*, and the semantic relation established between them in the proposition. In the final step (Step 4), the only direct core argument of the verb, *toe sorele* 'your sisters', is assigned the macrorole undergoer.[12] The sentence has been interpreted in full.

Starting from the consideration that the structure under discussion is a presentational construction, i.e., a construction which introduces an event into the universe of discourse (Section 1), and which does not provide an argument that can serve as the aboutness topic of the predication, we claim that another construal of the clitic *je* is possible. This interpretation is different from the locative reading in a subtle but significant way: *je* is not interpreted as the spell-out of the goal location of the event of arrival, but rather as a marker of the deixis of the discourse situation in which the event is announced and to which the event has relevance. The locative meaning of *je* plays a key role in this interpretation because the deixis of the discourse situation includes the spatio–temporal coordinates of the announcement. The locative deixis of the discourse situation and the goal location of the arrival can, of course, coincide. We should also add that the deixis of the discourse situation includes the speaker, who is the deictic centre of discourse (Vanelli 1972). Indeed, *je* can also indicate the spatio-temporal coordinates of the speaker at the time of the event, the speaker having experienced the event first-hand and providing a direct connection between the event and the discourse situation.

Adopting traditional terminology, in this construal *je* can be said to be—or to mark—the subject of predication. From our perspective, however, a subject of predication need not be a syntactic subject (or PSA). Indeed, *je* is not a syntactic subject in (37): it does not occur in a subject clitic position, nor does the inflected verb agree with it, as it agrees with the postverbal NP instead. In our analysis, the subject of predication is an aboutness topic, i.e., what the sentence, or utterance, is about (Gundel 1988). With specific respect to *je*, we claim that it is an index of the discourse situation, which is the aboutness topic of the utterance.

In Figure 4, we represent the syntax–semantics linking of (37), assuming *je* is an index of the discourse situation. The steps in the linking are the same as discussed above, with one important difference: since it encodes the spatio–temporal coordinates of the discourse situation, *je* is linked to the Discourse Representation Structure, which introduces these coordinates. The linking with the semantics of the verb is possible but not necessary, which explains why reanalysis can occur and why the syntax–semantics linking can be lost in diachrony. Note that the postverbal RP is interpreted as the PSA in Figure 4, i.e., the controller of verbal agreement, a grammatical relation which, in our framework, need not correlate with a specific position in syntax. This is in line with the theoretical assumptions on grammatical relations which are independently made in our framework (Section 4).

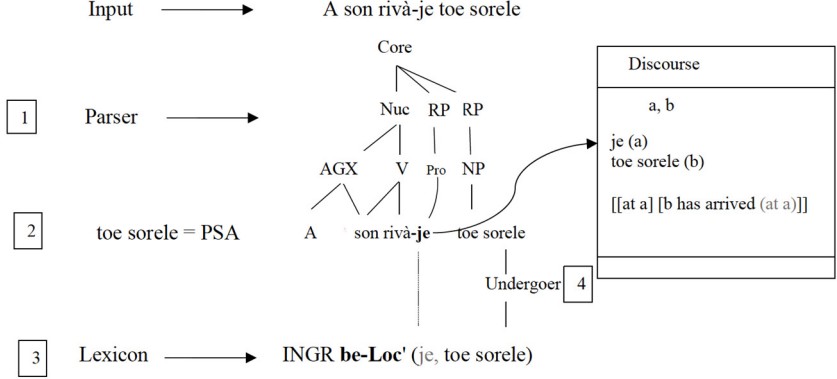

**Figure 4.** Pattern (iii) in syntax–semantics linking: interpretation of *je* in (37) as an index of the discourse situation.

It is this construal of *je* that can be extended to presentational constructions with verbs which do not describe motion. A relevant example is given in (38) (cf. 33b).

(38)     **A**     **son**     dimagrìje            tanti     cit.        (Turinese)
         SCL.3PL     be.3PL     lose.weight.PSTP.PRESCL     many     children
         'Many children have lost weight.'

The logical structure of *dimagrì* 'lose weight' does not comprise a locative argument. Therefore, unless a location is in the presupposition, in which case *je* will be a resumptive clitic, but (38) will not be an all-new utterance, *je* can only be understood as the marker of the deixis of the discourse situation in which the event is announced. We represent the latter alternative in Figure 5. The key point to note here is that *je* is solely interpreted through a direct linking from syntax to discourse. Thus, different interfaces are relevant to the interpretation of (38) compared with (37).

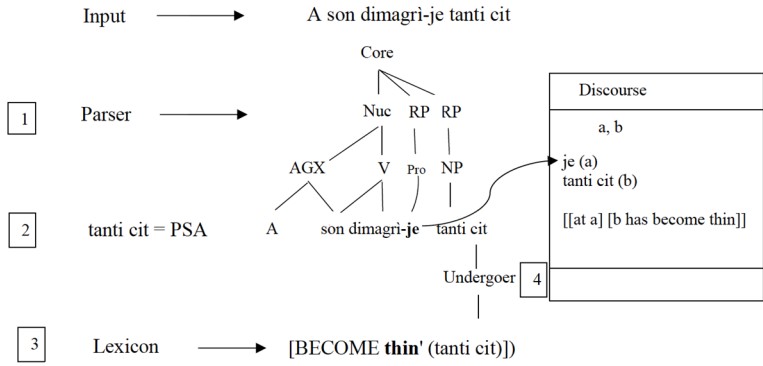

**Figure 5.** Pattern (iii) in syntax–semantics linking: interpretation of *je* in (38) as an index of the discourse situation.

The direct syntax–discourse linking ensures that the Completeness Constraint is satisfied and that the input is interpreted in full. While there is a notion of an Extended Completeness Constraint in RRG (Van Valin and Latrouite 2023) (Section 4), this notion has to date only been applied to cases of direct semantics–discourse, or discourse–semantics, linking, i.e., cases whereby predicates or arguments that are part of the semantics of the clause do not show up in its syntax (see, e.g., Shimojo 2008). In this article, we are applying the said notion to a direct linking which leaves out the semantic representation that is built from the Logical Structure of the verb (see Step 3 in Figure 5). This is no trivial matter in RRG, since in this framework meaning is assumed to be built compositionally, starting from the Logical Structure(s) of the predicator(s). It is therefore important to note that the insight that we are seeking to capture is not that meaning is built from syntax. Rather, our claim is that although meaning is built from the composition of Logical Structures that are stored in the Lexicon, utterances are fully interpreted within their discourse context. Following the view that there cannot be a topicless proposition, we assume that if a topic is not provided by the predicators in the clause, it will be provided by the discourse situation. The parallel architecture of RRG allows us to locate this topic where it belongs, i.e., in Discourse Representation.

### 5.2. Pattern (iii) vis-à-vis Patterns (i) and (ii)

As was mentioned, we take the pattern discussed in the previous section to have originated from pattern (ii), which exhibits V–S agreement but no presentational clitic. Indeed, neither pattern (ii) nor pattern (iii) were traditionally attested in Turinese (Burzio 1986; Parry 2013, among others). Pattern (ii) was then introduced because of pressure from Italian (Ricca 2008; Flecchia 2022), while pattern (iii) is the most recent one and has so far gone unnoticed. The comparative evidence from Milanese supports the hypothesis that

pattern (iii) derives from pattern (ii), since this dialect testifies to the stage that precedes the introduction of pattern (iii) into the system (see Table 2).

Let us briefly illustrate the syntax–semantics linking in patterns (i) and (ii). The linking in the former pattern is shown in Figure 6, which represents example (39) (cf. 23b).

(39)  **A**          **l'è**              **nassùje**              tante     fior.     (Turinese)
      SCL.3SG    AUXCL-be.3SG    be.born.PSTP.PRESCL    many     flowers
      'Many flowers have appeared.'

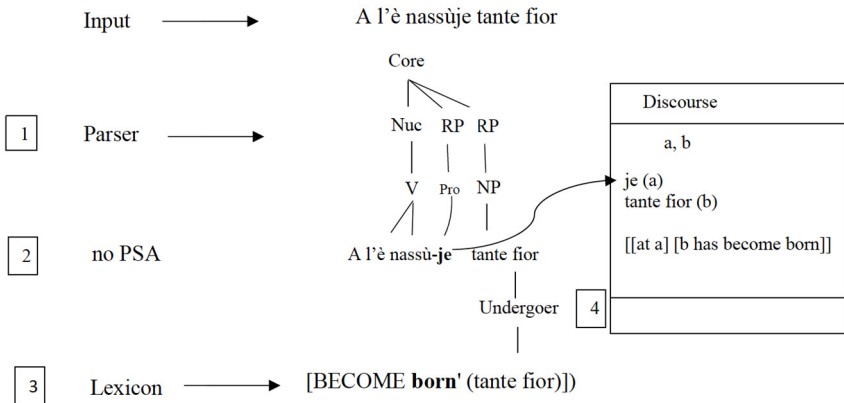

**Figure 6.** Pattern (i) in syntax–semantics linking: interpretation of *je* in (39) as an index of the discourse situation.

Not only is the auxiliary *è* 'is' unambiguously singular in Turinese, but *a* can be a third-person singular clitic (see note 4). The mismatch in agreement specifications between the auxiliary and the postverbal phrase is indicated by the absence of an agreement index (AGX) node in Figure 6. This mismatch results in the failure of PSA assignment, which is a key difference between the structure in (39) (Figure 6) and the one in (38) (Figure 5). The requirement of a topic is, however, satisfied by the direct syntax–discourse linking.

In our framework, the requirement of a presentational clitic in pattern (i), which is prevalent in Milanese and conservative in Turinese, can be considered to be an instruction of the presentational construction of these dialect varieties. The fact that these are subject–clitic dialects is relevant here. While in SV(O) topic-comment constructions, aboutness is expressed by S in concomitance with an agreeing subject clitic, depending on grammatical person, in VS presentationals, aboutness is expressed by a clitic alone. The autochthonous presentational construction thus has features that reflect a broader property of the grammar of these dialects.

The reverse situation is found in pattern (ii), illustrated in (40) and Figure 7. Here, a PSA is individuated, since *son* 'are' is unambiguously plural, like the postverbal phrase, and *a* can be interpreted as a third-person plural clitic, but the requirement of an aboutness topic is not satisfied overtly.

(40)  **A**          **son**          dimagrì              tuti     i        cit          (Turinese)
      SCL.3PL    be.3PL        lose.weight.PSTP    all      the     children
      ëd            la          scola.
      of            the         school
      'All the children of the school have lost weight.'

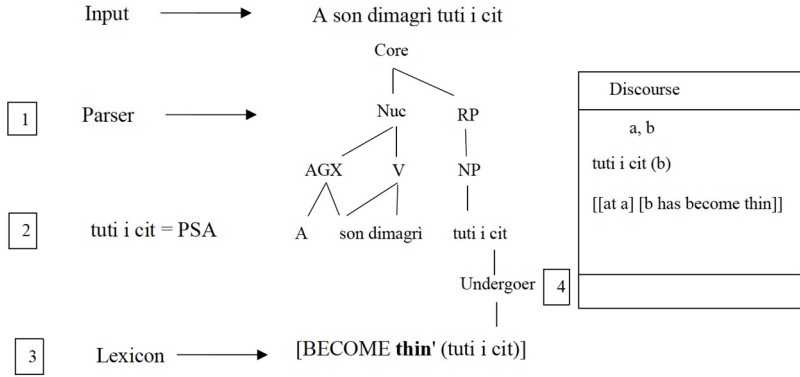

**Figure 7.** Pattern (ii) in syntax–semantics linking (cf. 40).

The postverbal RP is modified by the universal quantifier *tuti* 'all' in (40), and the restrictor *ëd la scòla* 'of the school' provides a frame (Lambrecht 1994, p. 90) for the interpretation of *tuti i cit* 'all the kids', whose referent is thus accessible to the interlocutors.[13] Indeed, we found that pattern (ii) was normally only deemed to be felicitous with definite postverbal noun phrases in Turinese. We should note that the same linking pattern is, however, in principle acceptable with indefinites, as can be seen in the Italian example in (41).[14]

(41)     Sono         dimagriti         tanti         bambini.         (Italian)
         be.3PL       lose.weight.PTCP  many          children
         'Many children have lost weight.'

In sum, the patterns analysed above demonstrate that the control verbal agreement, in our terms PSAhood, and the overt expression of an aboutness topic are in principle independent from each other. Pattern (i) is characterized by the latter but not the former (see Figure 6); in pattern (ii), we have a PSA, but not an overt expression of aboutness (see Figure 7); lastly, in pattern (iii), both aboutness and PSAhood are realized overtly, albeit separately (see Figures 4 and 5).

*5.3. Subject of Predication vis-à-vis Subject*

The analysis proposed in previous sections sheds light on the notion of subject of predication, treating it as orthogonal to that of controller of verbal agreement. The idea that the aboutness requirements on the clause are in principle separate from case and agreement, which, in other frameworks, can be satisfied within the verb phrase, is by no means new (Bianchi 1993; Cardinaletti 2004; Ojea 2017, etc.). In the past, however, the aboutness features of the clause have been associated with a subject position, regardless of the framework adopted in the analysis or the languages studied (Aissen 1999; Saccon 1993; Tortora 1997, 2014; Cardinaletti 2004; Parry 2013). In syntactic research, the debate has centred around the issue of whether and how this position is activated in all-new constructions (Cardinaletti 2004; Ojea 2017).

Couching our analysis in a parallel architecture framework, in this article we propose a change of perspective. We claim that, *qua* aboutness topic, the subject of predication can be an argument of the verb, which figures in the semantics and the syntax of the clause. However, it need not be an argument of the verb, in which case it is provided by the discourse situation and must be represented in discourse alone. The structures in which the subject of predication is an argument of the verb differ from those in which it is not in terms of the interfaces that are relevant to their interpretation: in the former type of structure, the subject of predication is interpreted through a direct linking between syntax and semantics (see Figure 3), which is not found in the latter type of structure (see Figures 5 and 6).

Against the backdrop of this analysis, pattern (i), which has an overt manifestation of the subject of predication, contrasts with pattern (ii), which does not, in terms of whether a linking from syntax to Discourse Representation is established in the interpretation of the sentence. The fact that only some of the languages of Italy require or allow this linking

is related to the expression of subjecthood in these languages, as has been argued in the relevant literature (Parry 2013) and as we pointed out in Section 5.2. However, we argue that this does not mean that the subject of predication is a subject. Rather, similarly to the spelling out of verbal agreement with a subject clitic, aboutness can also be marked by a clitic. Drawing on Bresnan and Mchombo (1987), Bentley (2018) suggested that this clitic expresses an anaphoric type of agreement, which is not internal to the clause or the verb phrase, but rather established between an anaphora and its antecedent in discourse. In fact, Bentley (2018) went as far as to claim that in presentational construction anaphoric agreement is in competition with the grammatical agreement between the verb and its argument. Whereas it now turns out, on evidence from contemporary Turinese (i.e., pattern (iii)), that anaphoric and grammatical agreement can both be marked overtly in presentational constructions, the connection between the morphological properties of the sister languages and the overt marking of aboutness is corroborated.

By way of conclusion of this section, we should mention and refute an alternative analysis of the data discussed in this article. It is in principle conceivable that *je* and cognates are mere markers of theticity, or of the presentational construction, without this being characterized by an aboutness topic of any sort.[15] If this were the case, the patterns discussed above would simply differ in the overt marking of theticity, while the development of the clitic would amount to a reanalysis from a locative pronoun to a theticity marker. Within the theoretical perspective of RRG, this analysis has a great deal of appeal, in that it does not require the postulation of a component of syntactic tree structure which has no correlate in semantics (see the arboreal representations in Figures 4–6) or the concomitant extension of the Completeness Constraint to cases of direct syntax–discourse linking. In addition, this analysis of *je* and cognates abides by Lambrecht's (1994) notion of sentence focus, traditionally adopted in RRG (see, e.g., Van Valin and LaPolla 1997, p. 207), which rules out a topical component.

Yet, the proposal expounded in this article is preferable on theoretical and empirical grounds. Applying Erteschik-Shir's (1997) insight that a predication is a function that maps a proposition to a topic, we rule out the notion that the interpretation of an utterance which occurs out of the blue should not start from an understanding of what the utterance is about. We claim instead that, in the absence of other clues, the utterance is understood to be about the discourse situation. Of course, the latter need not be encoded in syntax: this conclusion emerges from comparison of pattern (ii), which is the only pattern available in Italian, with patterns (i) and (iii), which we found in the dialects.

Our analysis does not violate any of the tenets of the RRG framework because the presentational clitic that is the output of the reanalysis discussed in Sections 3.2 and 5.1 is referential, its reference being in discourse and not in semantic representation. We should add that the most advanced RRG treatments of information structure do acknowledge that sentence focus is structured in a topic–comment articulation (Bentley 2023b). Therefore, our proposal constitutes another step in a direction which is already pursued within the framework.

As for the empirical advantages of the analysis proposed in this article, not only does it shed light on the microvariation attested within and across dialects, relating such variation to broader properties of the grammars of these dialects, but it can also serve as the starting point of a comparative analysis of presentationals and existentials. As was mentioned, in Italian and in some dialects of Italy, an etymologically locative clitic occurs in existentials (cf. 16) but not in presentationals (cf. 18). Since existentials are normally thetic, similarly to presentationals, this mismatch in the occurrence of the etymologically locative clitic suggests that theticity alone cannot capture the occurrence of the clitic. Further comparison of existentials and presentationals is thus needed, although it would obviously go beyond the scope of this article.

We thus propose an account of the etymologically locative clitic as a spell-out of the deixis of the discourse situation in which the presentational construction is interpreted, and we claim that the discourse situation is the aboutness topic of the all-new utterance. In

the spirit of RRG, we represent the discourse situation in Discourse Representation and we link the clitic directly from syntax to discourse.

## 6. Conclusions

Presentational constructions, i.e., constructions which introduce an event into the universe of discourse, raise the question of what it means for a predication to be new in information structural terms. Following a philosophical tradition established by Brentano and Marty (see note 15), such constructions have traditionally been thought to lack a topic (see Lambrecht's 1994 notion of sentence focus). However, some scholars have claimed that all predications require a subject (Bianchi 1993) or topic (Erteschik-Shir 1997), which, in the absence of co-textual or contextual clues, can be an understood, and usually speaker-oriented, spatio-temporal dimension (Benincà 1988; Calabrese 1992; Saccon 1992; see also Erteschik-Shir 1997).

The dialects of Northern Italy have figured prominently in this debate because many such dialects exhibit an etymologically locative clitic in presentational constructions, which clusters, or occurs in complementary distribution, with the subject clitics figuring in SV order. The presence of the etymologically locative clitic has traditionally been known to be mutually exclusive with verbal agreement with the postverbal argument, which thus fails to behave as a subject. Therefore, it has been claimed that the etymologically locative clitic of the presentational construction is a subject of predication, i.e., the marker of a spatio-temporal location provided by discourse and, at the same time, a subject agreement marker, comparable to the subject clitic of SV order (see, among others, Tortora 1997, 2014; Parry 2013).

In this article, we have brought to light primary evidence from Milanese and Turinese, two dialects spoken in the Northern Italian regions of Lombardy and Piedmont, respectively. We have noted that the variation attested in contemporary Turinese is more complex than has so far been noted in the literature, in that the etymologically locative clitic does not occur in a subject clitic position and is not mutually exclusive with V–S agreement, thus challenging the idea that it occurs in a subject position and marks agreement with a syntactic subject. The same evidence does not, however, challenge the view that this presentational clitic is the marker of a discourse aboutness topic.

We couched our analysis in a parallel architecture theory, Role and Reference Grammar (RRG), where the various modules of grammar (syntax, semantics, discourse) are represented independently of each other and can interact directly with each other. In our account, the presentational clitic can be a locative argument in the semantic representation of the predicate or the marker of the deixis of the discourse situation, which is an aboutness topic represented in discourse representation. In the latter case, the presentational clitic can be called a subject of predication. Crucially, our notion of subject of predication is orthogonal to that of syntactic subject and need not coincide with it. We have claimed that the disentanglement of the notions of subject of predication and syntactic subject does justice to the microvariation attested in the dialects of Northern Italy, which concerns the interfaces that are involved in the parsing of utterances and sheds light on the presentational construction itself, laying the foundations for a proper characterisation of its similarities and differences with existential constructions.

Importantly, the analysis pursued in this article considers discourse to be an integral part of grammar, an idea which has always been defended in RRG (Van Valin and LaPolla 1997; Bentley 2023b), and which is also shared by work of other theoretical persuasions (Lambrecht 1994; Erteschik-Shir 1997; Ojea 2017). Discourse is directly involved in the interpretation of utterances and in patterns of variation and processes of change, as evidenced by the reanalysis and consequent re-grammaticalization of the locative clitic of the dialects under scrutiny into a presentational clitic, which is a change in the interfaces that are relevant to the interpretation of the construction.

**Author Contributions:** Conceptualization, D.B. and F.M.C.; Data curation, D.B. and F.M.C.; Formal analysis, D.B.; Investigation, D.B. and F.M.C. All authors have read and agreed to the published version of the manuscript.

**Funding:** This research received no external funding.

**Institutional Review Board Statement:** Not applicable.

**Informed Consent Statement:** Informed consent was obtained from all subjects involved in the study.

**Data Availability Statement:** Questionnaire transcriptions are available from the authors upon request.

**Conflicts of Interest:** The authors declare no conflict of interest.

## Notes

[1]    The dialects of Italy are Romance languages, and hence daughters of Latin and not varieties of Italian, the major Romance language spoken in Italy. They are conventionally referred to as *dialects* because they have very little, if any, socio-political recognition. For further detail, including the classification of these languages into different subfamilies, we refer to Parry (1997) and Loporcaro ([2013] 2020).

[2]    In the glosses of the examples we use the Leipzig abbreviations, with the following additions: AUXCL = auxiliary clitic; DATCL = dative clitic; EXPL = expletive; LCL = locative clitic; OCL = object clitic; PARTCL = partitive clitic; PF = (existential) proform; PRESCL = presentational clitic; PSTP = past participle; SCL = subject clitic. We maintain the original glosses of the examples that are drawn from the secondary literature.

[3]    Subject clitics are found in northern Italian dialects and cannot indiscriminately be assumed to be subject agreement markers (Renzi and Vanelli 1983; Rizzi 1986; Brandi and Cordin 1989; Benincà 1983, 1994; Poletto 1993, 2000; Vanelli 1997; Cardinaletti and Repetti 2010; Poletto and Tortora 2016). It would, however, go beyond the scope of this article to consider the variation in subject clitics that occurs outside the presentational construction.

[4]    The Turinese form *a* is a third-person singular or plural subject clitic (Regis 2006a, 2006b; Tosco et al. 2023, pp. 177–79; Regis and Rivoira 2023, p. 43). We assume that it is singular in (3) and plural in (4), in accordance with the number agreement specifications on the perfect auxiliary *esse* 'be'. We should mention that in some Northern Italian dialects there is another *a* clitic, which characterizes presentational constructions and behaves differently from subject clitics (Benincà 1983; Bernini 2012; Vai 2020). A comparative analysis of this *a* and the *a* that marks lack of number agreement in Turinese (cf. 3) is desirable but beyond the scope of this work. Here, we follow Tosco et al. (2023, p. 184) in analysing Turinese *a* as a third-person clitic, including when it occurs in presentational constructions. As for the form *l'*, it is a dummy proclitic, required by the vowel-initial forms of 'have'/ 'be' (Brandi and Cordin 1981; Pescarini 2016, pp. 748–49; Tosco et al. 2023, pp. 261–62; Regis and Rivoira 2023, p. 55). Following a long-established tradition, we gloss it as AUXCL (auxiliary clitic), regardless of whether it precedes an auxiliary or a copula.

[5]    The questionnaire included 36 multiple-choice dialect entries, each preceded by contextual information. The interviews were conducted in two different stages. Author A interviewed two native speakers of Milanese in the period between November 2014 and June 2015 (see Author A XXX), while Author B interviewed nine Turinese speakers in the period between December 2022 and September 2023. The native speaker informants (five women and six men) were aged between 40 and 80 years. Their level of education ranged from *scuola media* 'middle school' to *scuola superiore* 'high school', with one exception: one of the Milanese informants had completed a university degree. They were all individuals who speak the dialect on a daily basis in informal contexts, that is, with family and friends. Unless otherwise stated, the examples that we will provide illustrate the one option (out of those given as multiple choices) that was selected as the preferred choice by all the speakers of the given dialect. While a larger and numerically balanced speaker sample would have been preferable, we note that speaker numbers are low for the two dialects under investigation, particularly in the city of Milan, where the first round of interviews was conducted. It is of course possible that the apparent homogeneity of the Milanese data is a mere side-effect of the small size of the sample. Nonetheless, this has no consequences for our analysis, which does not adopt quantitative methods or aim to capture each dialect exhaustively, but rather proposes an explanation of the microvariation that we attested.

[6]    We should note that, in the examples with the third-person plural pronoun, verb agreement and/or lack thereof are both deemed to be acceptable by some speakers.

[7]    *A* is glossed as EXPL(etive) in (19) and (21) to follow Parry (2013).

[8]    Again, we follow Parry's (2013) glossing conventions here.

[9]    A note on the early varieties of the Centre-South is in order. These were null-subject vernaculars and never lacked V–S agreement. Yet, the clitic also emerged and established itself as a component of the existential construction in these vernaculars. The view that the clitic was reanalysed as a subject agreement marker satisfying a syntactic subjecthood requirement does not capture its development into an existential proform in these vernaculars. Instead, evidence from a geo-linguistically varied corpus of early Italo-Romance texts suggests that the clitic appeared in VS copular structures to resume a distant topical locative phrase. This ensured that the conditions of discourse coherence and cohesion were met in the narrative. In existentials, the clitic became the marker of the implicit contextual domain of these constructions (Francez 2007). This view of the emergence of the existential

proform (see Ciconte in Bentley et al. 2015, pp. 248–49, 254–56) accounts for the variation in V–S agreement in all the early Italo-Romance varieties but does not conflict with the analysis of the clitic as an agreement marker in the northern vernaculars, where it came to be associated with a subject position.

[10] It is worth pointing out that in the early sources presentational VS constructions are consistently introduced by spatio–temporal adverbials (e.g., *allora* 'then', *adunc(a)* 'then/at that point', *donde* 'thereafter'/'therefore', etc.) derived from locative etyma (Ciconte 2018, pp. 141–42). In the logo-deixis of the written domain, where there cannot be an implicit reference to the communicative situation, these adverbials spell out a narrative aboutness topic, similarly to how *je* spells out a discourse aboutness topic in modern Turinese.

[11] It is worth pointing out that this constructional requirement is not valid in all dialects, as testified by the dialect of Grosio, where we found a fourth pattern without clitic or V–S agreement ({-Cl; -Agr}, cf. 18a) in a previous survey (Bentley 2018).

[12] Core arguments are arguments that are required by the Logical Structure of the verb, i.e., the semantic representation that is stored in the lexicon for the verb. Direct core arguments are unmarked, or marked by case alone, differently from oblique arguments, which are adpositionally marked.

[13] These aspects of the semantics of the noun phrase would be taken care of in its semantic representation, which we do not provide here for brevity.

[14] The definiteness contrast between Turinese and Italian reflects the microvariation in the constraints on PSAhood that are at work in the two languages, an issue which goes beyond the scope of this article (see Bentley 2018; Bentley and Cennamo 2022).

[15] This hypothesis was suggested to us by Jürgen Bohnemeyer at the 17th International Conference on Role and Reference Grammar (Heinrich Heine University of Düsseldorf, 14–16 August 2023). We refer the reader to Kuroda (1972) and Sasse (1987) for the thetic/categorical distinction, which originated in the work of the Swiss philosopher of language Anton Marty (1847–1914), who in turn developed ideas by the German philosopher Franz Clemens Brentano (1838–1917).

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
