# Peer review of "Microvariation at the Interfaces: The Subject of Predication of Broad Focus VS Constructions in Turinese and Milanese"

_languages, doi:10.3390/languages9020037_

Round 1

Reviewer 1 Report

Comments and Suggestions for Authors

Great article. Thorough, clearly written and advancing relevant descriptive and theoretical points.    

It would be interesting to have a little more discussion on the status of the Turinese sentence-initial "A" (Expletive/ 3SG). Since it is invariant, and it is not obvious that it has referential properties, I am not sure that it should be linked to the AGX node, originally proposed to deal with pronominal feature-bundles. (This also applies to the auxiliary esse, which could be linked to V, with only its pronominal features realized in the AGX).    

Regarding the interpretation of Turinese "je", in its non-locative uses it is variously treated as a "presentational clitic" (§3 and elsewhere), as "stage/narrative topic" (fn.8), as "aboutness topic" (p17) and as a "discourse topic" (p17 and Figure 5). Given its deictic properties, it seems to me that it could be best analyzed as a "stage topic" throughout, leaving the assessement of its possible interpretive/ discourse-functional nuances to a future study based on spontaneous conversational / narrative data.

The tabs on the examples need to be corrected. There is a typo on p.23. It reads: "Milanese and Turinese, two dialects spoken in the Northern Italian regions of Lombardy and, respectively, Piedmont." I think it should say "Milanese and Turinese, two dialects spoken in the Northern Italian regions of Lombardy and Piedmont, respectively". There are a couple of not listed references (Marty, Brentano).

Author Response

Please, find our responses in the attached PDF file

Reviewer 2 Report

Comments and Suggestions for Authors

I have  two major concerns about this paper.

1) the data collection part is not properly discussed. Some information is given in Footnote 5, but NO crucial information (such as age, sociolinguistic background) about the informants is given. Also the way in which data are presented is somehow misleading, since no information is given about the acceptance rate of the sentences, nor it is said whether the sentences have been accepted by all speakers.

2) The authors propose an account of data couched within "Role and Reference Grammar". I find that such approach should be better motivated: specifically, the author should explain why such approach is superior to the accounts couched within Generative Grammar which have discussed VS word orders in depth.

Minor comments:

1) the alignment of examples and glosses should be fixed throughout the paper;

2) Example 4: couldn't this A be a discourse marker of the type discussed for Paduan by Benincà?

Author Response

Please, find our responses to the attached PDF file
